# The Factors Associated with the Development of COVID-19 Symptoms among Employees in a U.S. Healthcare Institution

**DOI:** 10.3390/ijerph20126100

**Published:** 2023-06-10

**Authors:** Dania M. Abu-Alhaija, Paidamoyo Matibiri, Kyle Brittingham, Victoria Wulsin, Kermit G. Davis, Thomas Huston, Gordon Gillespie

**Affiliations:** 1College of Nursing, University of Cincinnati, Cincinnati, OH 45221, USA; matibipr@mail.uc.edu (P.M.); gillesgl@ucmail.uc.edu (G.G.); 2College of Engineering, University of Cincinnati, Cincinnati, OH 45221, USA; brittikt@mail.uc.edu; 3Department of Environmental and Public Health Sciences, College of Medicine, University of Cincinnati, Cincinnati, OH 45267, USA; wulsinvw@ucmail.uc.edu (V.W.); daviskg@ucmail.uc.edu (K.G.D.); 4College of Engineering and Applied Sciences, University of Cincinnati, Cincinnati, OH 45221, USA; hustontr@ucmail.uc.edu

**Keywords:** COVID-19, demographics, healthcare workers, occupational, symptoms, vaccination

## Abstract

Healthcare workers have experienced increased occupational health risks caused by COVID-19 disease. The purpose of this project was to examine the relationships between reporting COVID-19 symptoms by employees in a healthcare institution in the United States and employees’ demographics, vaccination status, co-morbid conditions, and body mass index (BMI). This project employed a cross-sectional design. It involved the analysis of data on COVID-19 exposure and infection incidents among employees in the healthcare institution. The dataset contained more than 20,000 entries. The results indicate that being female, African American, between 20 and 30 years old, diagnosed with diabetes, diagnosed with chronic obstructive pulmonary disease (COPD), or on immunosuppressive medicines is associated with greater reporting of COVID-19 symptoms by the employees. Furthermore, BMI is associated with reporting COVID-19 symptoms; the higher the BMI, the greater the likelihood of reporting a symptomatic infection. Moreover, having COPD, being 20–30 years old, being 40–50 years old, BMI, and vaccination status were significantly associated with employees reporting symptoms while controlling for other variables associated with reporting the symptoms among the employees. These findings may be applicable to other infectious disease outbreaks or pandemics.

## 1. Introduction

The severe acute respiratory syndrome coronavirus 2 (SARS-CoV-2) of 2019 (COVID-19) has imposed a great burden on healthcare systems worldwide. Frontline workers, who directly provide care for patients infected with COVID-19, were placed at an increased risk of exposure to and death caused by infection with the COVID-19 virus. A survey conducted by the Centers for Disease Control and Prevention (CDC) on COVID-19 infection rates in the United States between February and April 2020 showed that 19% of the respondents infected with COVID-19 were identified as healthcare workers. Of the infected healthcare workers, 73% were female, 6% were at least 65 years old, 38% indicated that they have at least one chronic condition, and 55% revealed that they contracted COVID-19 from patients at work [1]. According to Ali et al., some of the factors that contributed to placing healthcare workers at high risk of occupational exposure to the COVID-19 virus were the unavailability of COVID-19 testing and an inadequate understanding of the disease [2]. Particularly significant factors that imposed a high risk of exposure to the COVID-19 virus among healthcare workers are the shortage of PPE in healthcare institutions and the overall distribution of resources needed to manage the pandemic across the U.S. [3]. Additionally, the stressful work environment, heavy workload, and lack of time to rest may have contributed to an increased occupational risk of infection with COVID-19 among healthcare workers [4].

The physical health impacts caused by the COVID-19 infection devastated healthcare workers. The CDC found that more than 1 million healthcare workers have contracted the virus, and over 2400 healthcare workers have died due to COVID-19 infection in the United States [5]. Furthermore, many healthcare workers experienced severe symptoms and needed hospitalization [6]. Similar to the general population, some healthcare workers infected with COVID-19 experienced “long COVID-19”, where the clinical symptoms of the disease persist for periods longer than the typical period [6]. Havervall and colleagues found that healthcare workers who had mild infections and experienced long-term symptoms reported that these symptoms were exhausting and disturbing to both their personal and work lives [7].

In addition, several healthcare workers have complained of and suffered from mental health conditions and problems due to the COVID-19 pandemic. Examples of these mental health problems are anxiety, burnout, and posttraumatic stress disorder [6]. Several of these mental health issues were potentially caused by increased stressful situations at work, such as increased workload and worries of contracting the virus, and exacerbated by new work practices and requirements [6].

The COVID-19 vaccine was developed to help prevent infection and reduce the severity of the disease. The CDC recommends that the general population receive the primary series of one type of the three vaccines available in the U.S. and approved by the U.S. Federal Drug Administration [8]. Fortunately, the U.S. Advisory Committee on Immunization Practices recommended prioritizing vaccinating healthcare workers during the early phase of COVID-19 vaccine authorization [9]. To enhance vaccination uptake among healthcare workers, several healthcare institutions mandated that workers receive a COVID-19 vaccine as a condition of maintaining their employment [10]. Razzaghi and colleagues report that more than 80% of U.S. healthcare workers have received the primary series of the COVID-19 vaccine by 19 April 2022 [11]. Despite the protective effects of the COVID-19 vaccine, symptomatic COVID-19 infections have been reported by fully vaccinated healthcare workers [12]. By 30 April 2021, more than 10,000 COVID-19 vaccine breakthrough infections had been reported in the U.S. [13]. Breakthrough infections along with the emergence of virus variants have put healthcare workers, even those who are fully vaccinated, at a higher risk of occupational exposure to infection with the COVID-19 virus [12].

Several factors affect the severity and clinical presentation of COVID-19 infections among healthcare workers. According to Lin et al., healthcare workers who are African American, Asian, or older than 50 years are more likely to develop a severe COVID-19 infection that leads to death [14]. Other factors that affect the clinical manifestation of the disease are the presence of co-morbid conditions and vaccination status [15,16]. Given that symptomatic COVID-19 infection is associated with severe infection [17], it is important to identify healthcare workers who are at higher risk of developing symptomatic COVID-19 infection. This would support the efforts directed toward protecting healthcare workers against the COVID-19 infection and other future outbreaks or pandemics. The purpose of this project was to examine the relationship between reporting COVID-19 symptoms by employees in a U.S. healthcare institution and their demographic characteristics, vaccination status, co-morbid conditions, and body mass index (BMI).

## 2. Methods

This quality improvement project employed a cross-sectional design. It involved secondary analysis of previously collected data on COVID-19 exposure and infection incidents among the employees of a U.S. healthcare institution. The University of Cincinnati Institutional Review Board’s approval was obtained on 29 March 2022, and the project was determined to be a non-human subject research project.

### 2.1. Dataset Description

The dataset included data related to COVID-19 infection and exposure incidents for approximately 15,000 employees. The dataset contained more than 20,000 data entry points, indicating that at least 5000 employees contracted COVID-19 more than once. Each COVID-19 infection or exposure incident for the same employee was handled as a separate data entry point. The subject institution follows the CDC guidelines with regard to COVID-19 prevention and management practices. Pursuant to these practices, employees were required to report information related to COVID-19 infection, exposure, and relevant symptoms to the institution’s Employee Health and Safety Team. Employees diagnosed with COVID-19 were required to provide updates on their symptoms on a daily basis until they tested negative for the virus. Data collection instruments were built, and data were collected and stored in Research Electronic Data Capture (REDCap), a web-based platform used for collecting and storing data in clinical research [18,19]. The data were collected between March 2020 and March 2022.

The dataset contained no identifiers. The dataset included variables that provided data on:Employee demographics: age, biological sex, race, ethnicity, height, weight, and BMI;Employee COVID-19 testing: the date of the COVID-19 test, the location of the testing lab, the date of the results, the date when the employee was notified about the test result, and the date when the manager was notified about the result;Employee exposure to the COVID-19 virus: whether the employee identified an exposure situation to the COVID-19 virus and the source of exposure;Onset date of COVID-19 symptoms, if any;Date of last day worked or being unable to work either at home or onsite;Whether the employee worked from home while on restrictions due to COVID-19 or COVID-19-like symptoms;Vaccination status, the number of vaccine doses received, and vaccine manufacturer (Moderna, Pifizer, Astra-Zeneca-Oxford, Janssen);Whether the employee needed oxygen at home;Whether the employee has any co-morbid conditions (e.g., high blood pressure, diabetes, stroke, immunological disorders).

All the values for the date variables represent the number of days between the actual historic date and the date when the entry was completed in REDCap. The variables height, weight, BMI, the date of the COVID-19 test, the date of the COVID-19 result, the date in which the employee was notified about the result, the date in which the manager was notified about the result, the date of symptoms onset, and the date of the last day worked were analyzed as continuous measurements. All other variables in the dataset were categorical. Several meetings with the employee health and safety team helped interpret the data.

### 2.2. Analysis

The IBM SPSS Statistics (IBM Corporation, Armonk, NY, USA) program was used to conduct descriptive and inferential statistical analysis [20]. A Python script was used to correct issues in continuous variables that contained unstructured data by creating proxy variables. Proxy variables contain values that are appropriate to be used in the analysis and are approximate to the values in the original variables that contain unstructured data. Before starting the main analyses, the dataset was prepared. This preparation involved naming the variables according to SPSS rules, defining the variables’ types and levels of measurement, assigning a label for each variable, recoding the variables as needed, and evaluating missingness, normality, and outliers. Some missing values were attributed to the branching logic feature that was used in some surveys and to introducing new surveys into REDCap after the data collection period began. During the analysis, cases with missing values were deleted on a case-by-case basis. In short, outliers and bad data points were deleted before starting the analysis.

Descriptive analysis was used to summarize the variables; frequencies, means, medians, standard deviations, percentages, ranges, and minimum and maximum values were computed for the variables in the dataset. Graphs, charts, and tables were created to visualize the data and to depict data problems such as missingness, outliers, and deviation from normality [21]. The independent sample *t*-tests were used to assess the relationships between the categorical dependent variable representing whether the employee reported symptoms and the continuous variables. Chi-square tests of independence were used to investigate the relationships between the dependent variable representing whether the employee reported symptoms and the other categorical variables. Variables that had a statistically significant relationship with reporting COVID-19 symptoms were introduced into a logistic regression model to evaluate the individual significant relationship of each variable with the outcome (reporting symptoms). All the variables were introduced into the regression model at once; therefore, each independent variable controlled for all other variables included in the model in the interpretation of its relationship with the dependent variable [21].

### 2.3. Strategies to Protect the Data

Several protection measures were adopted to protect the data. First, the dataset was stored on a secure university research drive. The data could be accessed only through the password-protected investigator accounts. Access to this research drive remotely requires a virtual private network (VPN). In case of accidental data loss, a dataset backup is available for data stored on this drive. To minimize loss of confidentiality, the identifiers of the participants had been deleted from the data by the institution that owned the dataset before it was obtained. Furthermore, the actual dates of the date variables were withheld, and the values in the age variable were converted to age ranges to provide extra levels of confidentiality.

## 3. Results

### 3.1. Descriptive Statistics

Table 1 shows the descriptive characteristics of the respondents, their vaccination status, and their exposures to COVID-19. Of the respondents, 5276 (26.4%) were 20–30 years old, 6422 (32.1%) were 30–40 years old, and 3940 (19.7%) were 40–50 years old. By race, 14,935 (74.1%) of the respondents were white, 3234 (16%) were African American, and 855 (4.2%) were Asian. Approximately 78.9% (*n* = 15,910) of respondents were born female, and 4158 (20.6%) were born male.

High blood pressure was the predominant co-morbid condition among respondents (*n* = 2441, 12.1%). Other co-morbid conditions included diabetes (*n* = 863, 4.3%), chronic obstructive pulmonary disease (COPD, *n* = 1256, 6.2%), and immunosuppressive diseases (*n* = 282, 1.4%). The average BMI for respondents was 29.7 kg/m^2^ (SD = 7.7, maximum = 80.7 kg/m^2^, minimum = 13.1 kg/m^2^). Knowing that the normal range of the BMI is between 18.5 and 24.9 kg/m^2^ and that a BMI that is 30.0 kg/m^2^ or more indicates obesity [23].

In terms of exposure to the COVID-19 virus, 1726 (8.6%) reported a patient as the source of exposure. Approximately 13% (*n* = 2645) reported exposure from a coworker, while 26.7% (5379) reported household exposure. Most of the respondents (*n* = 10,539, 69.7%) reported experiencing COVID-19 symptoms. The vast majority of respondents (*n* = 12,283, 86.1%) reported being vaccinated against the COVID-19 virus.

### 3.2. Inferential Statistics

Table 2 presents cross-tabulations, chi-square results, and odds ratios of reporting COVID-19 symptoms by employee characteristics, co-morbid conditions, and vaccination status; only statistically significant relationships are reported. Significant relationships were found between reporting COVID-19 symptoms and respondents’ age, race, biological sex, co-morbid conditions (i.e., diabetes, COPD, being on immunosuppressive medicines), and vaccination status. Employees who were between 20 and 30 years of age, African American, female, diagnosed with diabetes, diagnosed with COPD, or on medicines that suppress immunity had higher odds of reporting COVID-19 symptoms compared to other employee groups. While employees who were between 40 and 50 years of age, identified their race as White, or those who received the COVID-19 vaccine had lower odds of reporting COVID-19 symptoms compared to other employee groups.

The results of *t*-tests indicate that there was a significant relationship between BMI and reporting COVID-19 symptoms (*t*(13708) = −6.72, *p* < 0.001). Employees who reported symptoms had a significantly higher BMI (M = 30.03, SD = 7.84) compared to those who did not report symptoms (M = 29.06, SD = 7.41).

Follow-up analyses were conducted to test the relationships between the employee groups that reported symptoms at significantly higher rates compared to other employee groups and their vaccination status (Table 3). There were significant relationships between receiving the COVID-19 vaccine and being aged 20–30 years, identifying oneself as African American, identifying oneself as White, being born female, or being diagnosed with CODP. Employees who were between 20 and 30 years of age, African American, female, or diagnosed with COPD had lower odds of being vaccinated against the COVID-19 infection compared to other employee groups. While employees who identified themselves as White had higher odds of being vaccinated against COVID-19 infection compared to other employee groups. There were no significant relationships between being aged 40–50, being diagnosed with diabetes, or being on medicines that suppress immunity and COVID-19 vaccination status.

### 3.3. Regression Model Testing

Table 4 presents the results of the logistic regression model testing. Hosmer and Lemeshow test results indicate that the model is a good fit for the data (X^2^ (8) = 15.12, *p* = 0.06). Each of the following variables was significantly associated with reporting COVID-19 symptoms when controlling for other variables included in the model: being 20–30 years old, being 40–50 years old, being diagnosed with COPD, vaccination status, and BMI, with being between 20–30 years old and BMI having the most significant relationship with reporting COVID-19 symptoms.

## 4. Discussion

The purpose of this project was to examine the association between reporting COVID-19 symptoms by employees in a U.S. healthcare institution and the employees’ demographic characteristics, vaccination status, co-morbid conditions, and BMI. The results of this project yielded information about the employee groups that had the highest risk of experiencing COVID-19 symptoms. This information can be used when creating COVID-19-related policies targeted to improve occupational safety practices, including processes and procedures to protect high-risk employee groups at healthcare institutions.

Identifying employee groups at higher risk of developing COVID-19 symptoms is an important infection prevention and control measure in healthcare institutions. Lin et al. found that specific COVID-19-related symptoms are indicators of illness severity and mortality among healthcare workers [14]. Furthermore, identifying these employee groups could help diagnose COVID-19 among employees at early stages through focused screening. Furthermore, it helps prevent the transmission of the virus, as the symptomatic presentation of COVID-19 infection is linked to a higher viral load [24,25].

The present study results indicate that being female, African American, between 20 and 30 years old, diagnosed with diabetes, diagnosed with COPD, or on immunosuppressive medicines was associated with greater reporting of COVID-19 symptoms by the employees. Furthermore, BMI is associated with reporting COVID-19 symptoms; the higher the BMI, the greater the likelihood of reporting a symptomatic infection.

Older adults are more likely to experience COVID-19 symptoms because adults acquire more co-morbid conditions with the advancement of age [25,26,27]. However, this study showed that employees who were between 20 and 30 years of age were more likely to experience COVID-19 symptoms compared to those from other age groups. Although evidence from the literature indicates that younger adults are less likely to experience COVID-19 symptoms (Poletti et al., 2021) [28], these results could be explained by the lower vaccination rates among employees between 20 and 30 years old compared to employees from other age groups. On the other hand, the employees who were between 40 and 50 were less likely to develop symptomatic COVID-19 disease. In this study, there was no significant association between this age group and being vaccinated against the COVID-19 virus. Therefore, this could be due to other factors such as having fewer close contacts and/or more social distancing among this age group compared to younger age groups, which comprised the majority of the employee respondents [29,30].

Evidence shows that African Americans are more likely to have a severe COVID-19 infection compared to Whites [31]. This supports the findings from the current study, in which African American employees were more likely to report COVID-19 symptoms. Importantly, employees from this group were less likely to report receiving the COVID-19 vaccine compared to employees from other racial groups. Research indicates that African American healthcare workers are disproportionally affected by COVID-19 disease [32]. Possible reasons could be higher rates of reusing personal protective equipment, insufficient access to personal protective equipment, higher rates of occupational exposure to the COVID-19 virus, and greater vaccination hesitancy by healthcare workers of color, including African Americans [32]. Furthermore, co-morbid conditions such as hypertension, asthma, diabetes, and obesity, which are linked to worse COVID-19 outcomes, are more prevalent among African Americans than Whites [31]. Except for that, it is possible that cultural influences on symptom perception and reporting could affect the discrepancy in reporting symptoms by employees from different racial groups in this study [33]. Although people from other minority groups are more likely to present with severe COVID-19 infection as well [31], this was not supported by this study’s findings and may be attributed to the underrepresentation of other racial groups among the employee respondents in this study.

The findings from the current study contradict those from Abate et al.’s meta-analysis regarding the relationship between biological sex and developing COVID-19 symptoms [34]. This study’s findings could be interpreted in light of the lower vaccination rates among female employee respondents. Another explanation is that, in the general population, females tend to report physical symptoms more often than males [35].

In agreement with this study’s findings, the reviews by Ejaz et al. and Al-Ani et al. indicate that diabetes, COPD, obesity, and some medications that suppress immunity are associated with severe and symptomatic COVID-19 disease [15,36]. Other health conditions such as hypertension, cardiovascular diseases, and renal diseases have been linked to severe and symptomatic COVID-19 disease as well [15,26]. Unexpectedly, no relationships between other co-morbid conditions and reporting COVID-19 symptoms were found among the employees in this study. This could be due to the effects of other personal, behavioral, or environmental factors that influence COVID-19 symptom development among these employee groups.

The COVID-19 vaccine has been found effective in preventing symptomatic COVID-19 infection [16]. Therefore, it is not surprising that employees in this study who received the COVID-19 vaccination were less likely to report symptoms compared to those who did not receive the vaccine.

The relationship with reporting COVID-19 symptoms became not significant for the variables: race, having diabetes, and being on medicines that suppress immunity, while controlling for other variables associated with reporting the symptoms among the employees. However, being 20–30 years old, being 40–50 years old, having COPD, vaccination status, and BMI remained significantly associated with reporting COVID-19 symptoms after adjustment; BMI and being between 20–30 years old have the most significant relationship with COVID-19 symptoms.

The current study results provided information on the employee groups that are at the highest risk of developing symptomatic COVID-19 infection at a U.S. healthcare institution. This would impact the policies and procedures for the continued fight against COVID-19 disease among the workers in the healthcare industry. In light of this study’s findings, strategies should be adopted by healthcare institutions to identify the employee groups who are at higher risk of developing severe COVID-19 infection. Once identified, these groups should undergo more frequent screening for COVID-19 infection and be provided with treatment plans that are tailored to their specific healthcare needs. Assigning healthcare workers to work on the frontlines should be done after careful consideration of their risk of severe infection. Equally important, more efforts should be directed toward addressing vaccination hesitancy among healthcare workers, who are at higher risk of symptomatic and severe infection. These results are applicable to other infectious disease outbreaks or pandemics.

## 5. Limitations

The data analyzed in this project was previously collected by a source independent of the project team. Therefore, the reliability and accuracy of this data cannot be verified. Moreover, the analysis was confined to specific variables in the dataset that were not sufficient to answer all the study questions. Another limitation is the presence of unstructured data in some of the variables, which hindered the ability to analyze these data in their original values. Creating new variables with proximate values resulted in the loss of some data. Additionally, most of the variables in the dataset were at the categorical level of measurement, which limited the ability to accurately quantify the relationships between variables. Moreover, the dataset included more than 20,000 entries. Thus, this large sample size might influence the significant results in this study, which could be clinically insignificant. Self-report bias is another limitation; the data were collected using self-report surveys, and there was no way to validate the responses provided by the employees. Further, causal relationships between the variables could not be assessed because the data were collected cross-sectionally. Another limitation is the large missingness in some variables due to introducing new surveys after data collection had started and due to the use of the branching logic feature when building the surveys in REDCap. Branching logic directed respondents to subsets of questions based on prior responses instead of allowing respondents to complete all questions. Additionally, although reporting COVID-19 exposure and infection incidents is required by the institution that provided the data, there is no information on the extent to which the employees complied with this policy or the exact number of employees who responded to the surveys. Lastly, the data that were analyzed in this project were collected in a single healthcare institution, which may limit the generalizability of the study results to broader populations.

Despite these limitations, this study provided insights on factors that are associated with the development of symptomatic COVID-19 infection among employees in the healthcare industry. Moreover, further testing was conducted to assess the relationship between these factors while controlling for others, which provided important information on the individual significance of these factors in the development of symptomatic COVID-19 infection.

## 6. Conclusions

Healthcare workers have been placed at an increased risk of exposure to illness and death caused by COVID-19. Several factors have been found to affect the severity and symptom development of the COVID-19 infection among healthcare workers. Identifying healthcare workers who may develop symptomatic COVID-19 infection could help detect the disease among them at earlier stages through more frequent and focused screening and prevent virus transmission. Moreover, strategies should be adopted by healthcare institutions to protect high-risk workers against COVID-19 disease.

## Figures and Tables

**Table 1 ijerph-20-06100-t001:** Descriptive statistics for the categorical variables *.

Variables	*n* (%)	Variables	*n* (%)
**Age** (**years**)		**Exposure identified**	
10–20	118 (0.6)	No	6688 (33.2)
20–30	5276 (26.4)	Yes	8426 (41.8)
30–40	6422 (32.1)	Total	15,114 (75)
40–50	3940 (19.7)	Missing	5051 (25)
50–60	2870 (14.3)	**Exposure type**	
60–70	1313 (6.6)	Person with COVID-19	9617 (49.5)
70–80	76 (0.4)	Not reporting exposure	2577 (13.3)
Total	20,015 (100)	No known exposure	5935 (30.5)
Missing	150 (0.7)	Under Investigation	1303 (6.7)
**Race**		Total	19,432 (100)
African American	3234 (16.0)	Missing	733 (3.6)
White	14,935 (74.1)	**Contact**	
Asian	855 (4.2)	Community contact	1462 (7.3)
Native American or Alaska Native	75 (0.4)	Patient Contact	1726 (8.6)
Native Hawaiian or Pacific	29 (0.1)	Household Contact	5379 (26.7)
Mixed Race	481 (2.4)	Coworker Contact	2645 (13.1)
Other	175 (0.9)	No known Exposure/contact	124 (0.6)
Prefer not to answer	622 (3.1)	Missing	9233 (45.8)
Missing	49 (0.2)	**Symptoms reported**	
**Sex**		No	4574 (30.3)
Born Female	15,910 (78.9)	Yes	10,539 (69.7)
Born Male	4158 (20.6)	Total	15,113 (100)
Prefer not to Answer	87 (0.4)	Missing	5052 (25.1)
Total	20,155 (100)	**Oxygen**	
Missing	10 (0)	Not on oxygen	13,773 (100)
**Co-morbid conditions**		Yes, on oxygen	9 (0.0)
High blood pressure **	2441 (12.1)	Total	13,782 (100)
Diabetes	863 (4.3)	Missing	6383 (31.7)
Chronic Obstructive Pulmonary Disease (COPD)	1256 (6.2)	**Vaccinated**	
Immunosuppressive diseases	282 (1.4)	No	1981 (13.9)
No high-risk diseases	9620 (47.7)	Yes	12,283 (86.1)
Heart Failure	178 (0.9)	Total	14,270 (100)
Immunosuppressive Medications	295 (1.5)	Missing	5895 (29.2)
Chronic Kidney diseases	96 (0.5)	**Number of vaccine doses**	
Heart attack	178 (0.9)	0	24 (1.8)
Abnormal Kidney lab results	96 (0.5)	1	55 (4)
Peripheral Artery Diseases	23 (0.1)	2	524 (38.2)
Aneurysms	10 (0.0)	3	558 (40.7)
Angina	178 (0.9)	4	185 (13.5)
Dialysis	96 (0.5)	5	23 (1.7)
Stroke or Transient Ischemic Attack	77 (0.4)	6	1 (0.1)
Heart block or heart diseases	178 (0.9)	Total	1370 (100)
Missing	6388 (31.7)	Missing	18,795 (93.2)

* Total number of responses was not provided for variables with multiple responses allowed. ** High blood pressure is 130 mmHg or greater for systolic blood pressure or 80 mmHg or greater for diastolic blood pressure [22].

**Table 2 ijerph-20-06100-t002:** Cross-tabulations of the employees reporting symptoms by employee characteristics, co-morbid conditions, and vaccination status.

Employee Variables	Symptoms Reported		Chi-Square	*p*-Value	OR *	95% CI (OR) **	
	Yes	No				LL	UL
**Age 20–30**							
20–30 years old	2821	1098	12.45	<0.001	1.16	1.07	1.25
Other age	7635	3435			1.00		
**Age 40–50**							
40–50 years old	2064	983	7.39	0.007	0.89	0.82	0.97
Other age	8392	3550			1.00		
**African American race**							
African American race	1695	632	12.57	<0.001	1.20	1.08	1.32
Other race	8844	3942			1.00		
**White race**							
White race	7760	3482	10.42	0.001	0.88	0.81	0.95
Other race	2779	1092			1.00		
**Biological sex**							
Female	8389	3544	8.87	0.01	1.14	1.04	1.24
Male	2104	1010			1.00		
**Diabetes**							
Yes	635	228	6.41	0.01	1.22	1.05	1.43
No	9904	4346			1.00		
**COPD**							
Yes	945	310	20.08	<0.001	1.36	1.19	1.55
No	9594	4264			1.00		
**Immunosuppressive medication**							
Yes	223	72	4.89	0.03	1.35	1.03	1.77
No	10,316	4502			1.00		
**COVID-19 vaccinated**							
No	1445	467	16.33	<0.001	1.26	1.13	1.41
Yes	7756	3158			1.00		

* OR = odds ratio; ** CI = confidence interval.

**Table 3 ijerph-20-06100-t003:** Cross-tabulations of high-risk employees reporting receiving COVID-19 vaccination by employee characteristics, co-morbid conditions, and vaccination status.

Employee Variables	Vaccinated		Chi-Square	*p*-Value	OR	95% CI (OR)	
	Yes	No				LL	UL
**Age 20–30**							
20–30 years old	3025	595	27.26	<0.001	0.76	0.68	0.84
Other age	9179	1367			1.00		
**African American race**							
African American race	1766	478	122.37	<0.001	0.53	0.47	0.59
Other race	10,517	1503			1.00		
**White race**							
White race	9218	1318	64.07	<0.001	1.51	1.37	1.68
Other race	3065	663			1.00		
**Biological Sex**							
Born Female	9670	1692	47.04	<0.001	0.63	0.55	0.72
Born Male	2613	289			1.00		
**COPD**							
Yes	920	204	18.53	<0.001	0.71	0.60	0.83
No	11,363	1777			1.00		

**Table 4 ijerph-20-06100-t004:** Regression model for the relationship between reporting symptoms by the employees and employee characteristics, co-morbid conditions, vaccination status, and BMI.

Variables	B	S.E.	Sig.	Exp (B)
Age 20 to 30	0.18	0.05	<0.001	1.20
Age 40 to 50	−0.12	0.05	0.02	0.89
African American	0.003	0.08	0.97	1.00
White	−0.08	0.07	0.22	0.92
Sex (Female)	−0.15	0.33	0.64	0.86
Sex (Male)	−0.20	0.33	0.55	0.82
Diabetes	0.09	0.09	0.29	1.10
COPD	0.20	0.07	0.01	1.22
Immunosuppressive medicines	0.26	0.15	0.09	1.30
BMI	0.02	0.003	<0.001	1.02
COVID-19 Vaccinated	−0.18	0.06	0.002	0.84
Constant	0.82	0.34	0.02	2.27

## Data Availability

Not applicable.

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
