# Peer review of "The Factors Associated with the Development of COVID-19 Symptoms among Employees in a U.S. Healthcare Institution"

_ijerph, 2023, doi:10.3390/ijerph20126100_

Round 1
Reviewer 1 Report
I would like to thank you for the effort in writing this article, although I think some changes are needed to improve the quality of the information you are trying to convey.
1.- Please, it is necessary to comment in the introduction on the particular situation experienced in the USA regarding, for example, the non-use of masks or characteristics of epidemic management, since the situation in the USA is different from that of other countries. And specify the situation in the health institution I suppose that the situation in your institution is different as other work centers.
2.- It is very important to specify in limitations that your results cannot be extrapolated to other work centers, because the sample is small and is not representative of the population.
Reviewer 2 Report
There is a lot of redundant information, e.g. lines 18-184 and Table 1 (race, sex) or lines 242-254. On the other hand,regarding the comorbidities, almost one third of employees were presented with missing data. Any further analysis would be biased. Also, redundant information is present in the text of Results.The study was conducted between March 2020 and March 2022. The vaccination status is mentioned globally, and there is no information regarding breakthrough infections or timing. Moreover, in Table 1 there are inconsistent data between the list of vaccinated (yes, no, missing 29%!!!!) and the number of doses starting with 0 doses and missing data in 93%. Lines 199-202 – it is improbable that employees worked with COVID-19 symptoms 18 days on average and a maximum of 366 days. It would be a huge lack of medical education and neglected preventive measures. Personal protective equipment is mentioned in the dataset but no further information is presented. Please reconsider all the results and reduce the text if the data is correctly presented in the tables. The conclusion is not entirely correct because healthcare workers were prioritized for vaccination – lines 404-405.
Round 2
Reviewer 2 Report
Comment: The study was conducted between March 2020 and March 2022. The vaccination status is mentioned globally, and there is no information regarding breakthrough infections or timing.
The answer: "we have talked about this in the discussion including breakthrough infections and timing. the authors talked about". Discussion should include comments upon the study results not only reference data (lines 310-320).
Author Response
Thank you for your comment!
In responding to this comment, we moved the information containing reference data to the introduction section.